# Immunological Mechanisms in Inflammation-Associated Colon Carcinogenesis

**DOI:** 10.3390/ijms21093062

**Published:** 2020-04-26

**Authors:** Takehiro Hirano, Daisuke Hirayama, Kohei Wagatsuma, Tsukasa Yamakawa, Yoshihiro Yokoyama, Hiroshi Nakase

**Affiliations:** Department of Gastroenterology and Hepatology, Sapporo Medical University School of Medicine, Minami 1-jo Nishi 16-chome, Chuo-ku, Sapporo 060-8543, Japan

**Keywords:** colorectal cancer, inflammatory bowel disease, ulcerative colitis, colitis-associated cancer, signaling pathway

## Abstract

Patients with chronic inflammatory bowel diseases are at an increased risk of developing colitis-associated cancer (CAC). Chronic inflammation positively correlates with tumorigenesis. Similarly, the cumulative rate of incidence of developing CAC increases with prolonged colon inflammation. Immune signaling pathways, such as nuclear factor (NF)-κB, prostaglandin E2 (PGE2)/cyclooxygenase-2 (COX-2), interleukin (IL)-6/signal transducer and activator of transcription 3 (STAT3), and IL-23/T helper 17 cell (Th17), have been shown to promote CAC tumorigenesis. In addition, gut microbiota contributes to the development and progression of CAC. This review summarizes the signaling pathways involved in the pathogenesis following colon inflammation to understand the underlying molecular mechanisms in CAC tumorigenesis.

## 1. Introduction

Inflammation is a physiological response that causes injured tissues to heal. The inflammatory process begins with the secretion of biomolecules from damaged tissues. Subsequently, white blood cells migrate to the injured tissue to rebuild tissue and help repair the injury. The inflammatory signaling cascade ends once the wound has healed. Chronic inflammation results from the activation of signaling pathways without being stimulated by injury. However, the mechanism(s) involved in sustained inflammation remain to be understood. Chronic inflammation functions in various steps involved in tumorigenesis, including cellular transformation, survival, proliferation, invasion, angiogenesis, and metastasis. Patients with chronic inflammatory bowel diseases (IBD), such as ulcerative colitis (UC) and Crohn’s disease (CD), have an increased risk of colorectal cancer (CRC) [1].

CRC is the fourth most common and third most commonly diagnosed cancer in the world, comprising 11% of all cancer diagnoses. More than 70% of CRC cases are sporadic. Hereditary genetic mutations resulting in Lynch syndrome, familial adenomatous polyposis (FAP), and Peutz–Jegher’s syndrome are associated with the development of CRC. Dietary habits including food components could induce changes in the composition of the gut microbiome and inflammatory microenvironment, which are also associated with initiation and progression of CRC [2].

Recent developments in therapeutic strategies against UC have been remarkable and result in a reduction in the requirement for surgery in IBD patients resistant to conventional therapy. However, the risk of colon cancer increases in patients with prolonged UC. A meta-analysis has recently shown that the cumulative incidence of colitis-associated cancer (CAC) is 0.1%, 2.9%, and 6.7% after 10, 20, and 30 years, respectively [3]. UC patients who develop advanced CRC have a poor prognosis and higher proportion of mucinous carcinoma and signet-ring cell carcinoma than sporadic CRC patients [4,5,6]. Therefore, surveillance endoscopy is an important diagnostic tool for patients with UC (especially those with long-standing disease).

Unlike with UC, the risk of developing CAC in patients with CD is not fully understood. Canavan et al. performed a meta-analysis to report a 2.9%, 5.6%, and 8.3% cumulative rate of CRC development after the onset of CD over 10, 20, and 30 years, respectively [7]. A recent population-based cohort study showed an increased risk of CRC-induced death in CD patients (adjusted for tumor stage) compared to healthy individuals diagnosed with CRC [8]. Thus, the prognosis of CRC is poor in patients with CD.

Multiple genetic changes in the cells of the intestinal mucosa are associated with the development of CAC from inflamed cells in patients with IBD. Mutations in the tumor-related genes of intestinal epithelial cells and organ systems can be attributed to the action of inflammatory cytokines and reactive oxygen species. Moreover, epigenetic changes and altered levels of microRNAs enhance inflammation, epithelial cell regeneration, and the development of CAC (Figure 1). This review focuses on the mechanisms involved in the immune responses and signaling pathways in inflammation-associated colon tumorigenesis.

## 2. NF-κB Pathway

NF-κB is a family of transcription factors. The mammalian NF-κB family comprises five members: p50, p52, p65 (RelA), c-Rel, and RelB [9]. NF-κB signaling includes the canonical and non-canonical pathways. In both pathways, NF-κB forms a complex with the inhibitor IκB in the cytoplasm. The IκB kinase (IKK)α, IKKβ, and IKKγ complexes regulate IκB degradation using the ubiquitin–proteasome system. The canonical pathway begins with the nuclear shuttling of NF-κB (p50-RelA) and degradation of IκB in response to various stimuli, including cytokines, growth factors, mitogens, microbial components, etc., which regulates the gene expression of multiple target genes [10]. In contrast, the non-canonical pathway is selectively involved in the development of lymphatic organ [11]. The tumor necrosis factor receptor superfamily members lymphotoxin-β receptor [12], CD40 [13], RANK, and B cell activating factor receptor [14,15] function as effectors in this pathway. Activation of the non-canonical NF-κB pathway does not involve IκBα degradation but relies on the translocation of p52-RelB into the nucleus along with the degradation of the NF-κB2 precursor protein, p100/RelB (Figure 2). Functionally, canonical NF-κB is involved in almost all the immune responses, while the non-canonical arm functions as a supplementary signaling axis that cooperates with the canonical NF-κB pathway in regulating specific adaptive immune responses.

NF-κB is a core regulator of inflammatory responses and is involved in the pathogenesis of numerous inflammatory diseases, including IBD. This pathway induces the expression of pro-inflammatory cytokines [16] and contributes to inflammation-associated tissue injury [17]. Furthermore, NF-κB is involved in tumor proliferation and survival [18] and regulates tumor progression and metastasis by promoting the induction of angiogenesis-related genes [19,20].

In-vivo mice experiments have demonstrated the role of canonical NF-κB signaling in the development of CAC. Azoxymethane (AOM)/dextran sulfate sodium (DSS)-induced CAC cells decrease in mice with IKKβ deficiency as compared to in wild-type (WT) mice. Moreover, the size of CAC tumors and pro-inflammatory cytokines in the intestinal mucosa significantly decreased in AOM/DSS-treated mice deleted for myeloid-specific IKKβ compared to in WT mice. Thus, the upregulation of intestinal NF-κB signaling stimulates the development of CAC, and activation in myeloid cells leads to tumor progression by modulating tumor properties and microenvironment [21]. Canonical and non-canonical NF-κB signaling are involved in the development of CAC. Nod-like receptor (NLR) 12 (NLR Family Pyrin Domain Containing 12), a member of the NLR family of intracellular sensors of molecular danger signals, is a negative regulator of non-canonical NF-κB signaling. NLRP12 knockout (KO) mice (with activated non-canonical NF-κB signaling) treated with AOM/DSS exhibit severe colitis and a higher proportion of CAC cells than WT mice [22].

The association between p53 and NF-κB has been examined [23]. Decreased p53 stability activates NF-κB and increases pro-inflammatory cytokine levels, such as IL-6 and IL-8, in DSS-treated mice with CAC. This triggers p53 in the progression of adenomatous lesions to invasive carcinoma in patients with IBD [24].

Among the pro-inflammatory cytokines, studies have focused on the role of TNF-α in autoimmune diseases: anti-TNF-α therapy remarkably decreases the clinical manifestation of diseases associated with inflammation [25]. TNF-α induces a variety of cellular responses by interacting with two transmembrane receptors, the 55 kDa TNF receptor type I (TNFR1) and 75 kDa TNF receptor type II (TNFR2) [26]. Under normal physiological conditions, TNFR1 is ubiquitously expressed in various cell types and tissues, whereas TNFR2 is predominantly expressed at low levels in immune and endothelial cells. TNFR1 activation can be induced by soluble or membrane-bound TNF-α, while TNFR2 is predominantly activated by membrane-bound TNF-α. Studies using TNFR KO mice and cell cultures have shown that pro-inflammatory and apoptotic pathways are activated by TNF and linked to tissue injury; tissue repair and angiogenesis are usually dependent on TNFR1 and TNFR2 signaling.

TNF-α antagonists reduce CAC growth in AOM/DSS-treated mice; the number of CAC cells was seen to decrease in TNFR1 KO mice compared to in WT mice [27]. Moreover, mice with DSS-induced colitis exhibit TNFR2 overexpression in the colon epithelial tissues, indicating the role of TNFR2 signaling in epithelial regeneration [28].

## 3. PGE2/COX-2 Pathway

Cyclooxygenase (COX) signaling is associated with tumorigenesis [29]. Isotypes of COX include COX-1 and COX-2. COX-1 is involved in organ homeostasis and is ubiquitously expressed in platelets, gastrointestinal tract, and kidneys, whereas COX-2 is expressed only in response to certain stimuli. Pro-inflammatory cytokines induce the expression of COX2 and PGE2, involved in vasodilation, thereby leading to inflammation. The arachidonic acid cascade involves the metabolism of COX-1 and COX-2 to prostaglandin G2 followed by prostaglandin H2. Multiple studies have shown the involvement of COX-2 in the development of colitis and CAC.

Mice with null mutations of COX-2-encoding gene (Ptgs2) or mice treated with COX-2 inhibitor show a reduction in the frequency of tumors in *Apc*^Δ716^ KO mice (model for FAP) [30,31]. Peroxisome proliferator-activated receptor δ (PPARδ) is implicated in COX-2/PGE2-associated CAC carcinogenesis. PPARδ activates PGE2 via PI3K-AKT signaling in tumor cells, thereby increasing adenomatous lesions in the colon in *Apc^Min/+^* mice. Notably, PPARδ inactivation reduces colon inflammation and adenoma formation in DSS-treated *Apc*^Min/+^ mice [32]. COX-2 is also involved in induction of BCL-2 expression, which decreases cellular apoptosis and tumor proliferation [33]. PGE2 directly enhances the expression of CXCL1 in human CRC endothelial cells. CXCL1 is a chemokine that activates and recruits neutrophils and induces angiogenesis [34], and its receptor, CXCR2, induces infiltration of myeloid-derived suppressor cells in CAC mice. Myeloid-derived suppressor cells accelerate tumor growth by suppressing the cytotoxic activity of CD8^+^ T cells, indicating their role in cancer immune evasion [35].

## 4. IL-6/STAT3 Pathway

IL-6 is produced in an NF-κB-dependent manner in innate immune cells within the lamina propria in response to intestinal injury, thereby regulating the survival and proliferation of intestinal epithelial cells (IECs). Thus, IL-6 is important in tissue homeostasis and regeneration. IL-6 is also critical for T cell survival, differentiation, and T-cell-dependent autoimmune disorders, including IBD.

IL-6 binds to soluble or membrane-bound IL-6 receptor (IL-6Rα) polypeptides, which interact with membrane-associated gp130 [36]; this triggers the activation of Janus kinases (JAK) and downstream effectors, signal transducer and activator of transcription-3 (STAT3) [37], Shp-2-Ras [38], and phosphatidyl inositol 3′ kinase (PI3K)-Akt [39]. STAT3 binds to specific DNA sequences and regulates the transcription of regulators of cellular proliferation (cyclin D1, proliferating cell nuclear antigen), survival (BCL-xL, survivin), and angiogenesis (VEGF) [40].

IL-6 stimulates the proliferation of IECs and regulates tumorigenesis in AOM/DSS-treated mice [41]. Interestingly, epithelial STAT3 is essential for the progression of colonic tumor by coordinating immune cell recruitment via the sphingosine-1-phosphate receptor (S1PR1). Upregulation of S1PR1 triggers the stimulation of tumor-infiltrating macrophages and dendritic cells and increases IL-6 concentrations in CAC. Thus, S1PR1 is an effector of inflammation-associated tumorigenesis [42,43].

Serum levels of IL-6 correlate with tumor size, staging, and metastasis in human CRC [44]. Active IBD patients exhibit increased levels of IL-6 and its receptors in the lamina propria. The activation of the IL-6/IL-6R axis and STAT3 in the intestinal mucosa of IBD patients induces the expression of antiapoptotic genes Bcl-2 and Bcl-xL [45]. IL-6 also mediates the regeneration of epithelial cells. The IL-6/STAT3 axis is involved in the expression of IL-22 [46]. IL-22 is a cytokine that is primarily synthesized by CD4^+^ T cells, natural killer cells, and NKT cells. The IL-22 receptor is expressed on epithelial cells in the skin, respiratory tract, and gastrointestinal tract [47] and plays an important role in the mucosal defense system by restoring the goblet cells and mucus layer [48]. Taken together, the IL-6/STAT3 axis has complex and diverse roles in intestinal cells [49].

## 5. IL-23/Th17 Pathway

IL-23 is produced by dendritic and other antigen-presenting cells and is a member of the IL-12 family of cytokines. IL-23 is composed of two subunits, p19 and p40, and its receptor comprises two subunits, IL-12Rβ1 and IL-23Rα [50,51]. IL-23 receptors (IL23Rs) have been implicated in chronic inflammatory diseases owing to their role in the differentiation of helper T cells (Th17). IL-23 enhances the levels of PGE2 and Th17 cell function [52]. Th17 cells produce inflammatory cytokines, such as IL-17A, IL-17F, IL-21, and IL-22 [53]. Patients with IBD exhibit increased levels of IL-17 in the serum and colon mucosa [54]. The IL-23/Th17 pathway is strongly involved in the pathophysiology of IBD and CAC [55]. In APC-deficient mice, IL-23 and IL-17A were upregulated in CAC as compared to that in the normal colon mucosa. Mice deficient in IL-23 or IL-23R showed decreased expression of pro-inflammatory cytokines, including IL-17A, in the colon mucosa and reduced cancer growth [56]. Mice with CAC depleted of the basic leucine zipper transcription factor ATF-like (Batf; regulator of Th17) exhibit reduced size of tumors and number of tumor cells. These mice contained low levels of IL-23 and upregulated IL-17A in the colon mucosa; tumor formation and intratumoral IL-23 expression were observed to be restored in environments with overexpression of IL-6 [57]. The NADPH oxidase p47PHOX upregulates IL-23 but downregulates IL-12 (regulator of Th1 differentiation) in myeloid dendritic cells [58]. Administering anti-IL-17A antibodies suppresses the development of CAC in 1,2-dimethylhydrazine/DSS-treated mice [59]. Additionally, AOM/DSS-treated IL-17A-deficient mice with CAC show reduced tumor development [60]. These data suggest the involvement of IL-23/IL-17 signaling in tumor development.

## 6. Purinergic Signaling

It is reported that purinergic pathways contribute to homeostasis in the intestinal immune system. Purinergic receptors contain adenosine (P1) and ATP (P2X) receptors, and their signaling pathway affects immune responses and inflammation, but also cell proliferation, cell differentiation and cell death [61]. In IBD patients, P2X_7_ receptors are upregulated in epithelial layer and mononuclear cells in lamina propria. P2X_7_-receptor-deficient mice showed a decrease in inflammatory cytokines and regulatory T cell (Tregs) accumulation in the colon [62,63]. Contrary to anti-inflammatory effects, blockade of P2X_7_ receptor increased tumor incidence in AOM/DSS-treated mice. The effect of tumor growth in this model mouse was associated with an increase in TGFβ1 expression, which influence immunosuppressive microenvironment and the direct stimulation of epithelial cell proliferation [64]. 

## 7. CAC and Gut Microbiota

Many studies have demonstrated the importance of gut microbiota in the development of CAC. Germ-free environments suppress the development of a variety of intestinal tumors in mice (such as IL-10-deficient and T cell receptor β chain/p53 double-deficient mice) [65,66]. Several studies have reported the inhibition of inflammation and tumorigenesis by probiotics [67,68], while some have shown that probiotics enhance tumorigenesis [69].

IBD patients possess increased numbers of the *Enterobacteriaceae* family and a decrease in *Firmicutes* [70]. Metagenomic and metabolomic analyses have revealed that the composition of gut microbiota depends on CRC staging [71]. However, there is little known about the gut microbiota in patients with CAC as compared to those in sporadic cancer (SC) patients.

Bacterial stimulation of toll-like receptors activates NF-κB, which triggers tumor development and growth [72]. AOM/DSS-induced colitic cancer is suppressed in TLR-4-deficient mice [73]. Mice carrying heterozygous mutations for APC, the gene responsible for FAP (*APC^Min/+^* mice), spontaneously develop CRC, whereas MyD88-deficient mice exhibit reduced tumorigenesis. Moreover, MyD88-deficiency in *Apc^Min/+^* mice strongly reduces the formation of benign polyps [74].

Studies on CAC models have shown that microbiota can promote or suppress colitis and tumorigenesis. *Bifidobacterium lactis* inhibits NF-κB in IECs, which prevents the development of acute colitis and CAC in mice [75]. It has been reported that the protease toxin produced by *Bacteroides fragilis* is an anaerobic bacterium, which causes enteritis and further develops into cancer. *B. fragilis*-induced inflammation and tumorigenesis involves the induction of Th17 cells by activating STAT3 [76]. Richard et al. reported comparison data on the mucosa-associated microbiota in patients suffering from CAC and SC and healthy subjects [77]. The gut microbiome in patients with CAC was enriched in members of the *Enterobacteriacae* family and *Sphingomonas* genus and a decrease in members of the *Fusobacterium* and *Ruminococcus* genus as compared to those in patients with SC. There was a difference in gut microbiota in patients with SC and adjacent normal colonic mucosa; however, patients with CAC did not exhibit similar phenotypes. CAC tumors possess an increased proportion of members of *Streptococcus* compared to in the surrounding healthy mucosa.

Several reports have shown the association between CAC and fatty acid metabolism. Short-chain fatty acids (acetate, propionate, and butyrate) and dietary fiber metabolites alleviate the symptoms of DSS-induced colitis [78]. Short-chain fatty acid transporters (MCT1, SMCT1) are expressed in the colonic epithelium and their expressions are suppressed in a DSS-induced CAC model. Thus, short chain fatty acid transporters have a protective role in patients with UC and colon carcinogenesis [79]. Western-style diet (high content of long chain fatty acids) promotes DSS-induced inflammation and accelerates the infiltration of macrophages, thereby leading to the development and progression of colon cancer [80].

Bacteria exert direct effects on tumorigenesis. The attachment and effacement of *Escherichia coli* suppresses the expression of mismatch repair proteins [81]. IBD is associated with reduced counts of *Clostridium* that plays a key role in the induction of Tregs [82]. Butyric acid also plays an important role in the induction of Tregs in the large intestine [83]. Interestingly, the number of Foxp3^+^ Tregs in the tumors of AOM/DSS-treated mice with CAC were higher than those in WT mice [84]. Over the years, tumor cells have been demonstrated to evolve multiple complex mechanisms to escape immune surveillance. To that extent, the following problems need to be elucidated: (1) origins of tumor-infiltrating Tregs; (2) conversion of conventional CD4 T cells to Tregs in the tumor; and (3) recruitment of Tregs to tumor sites. Thus, studies on the involvement of Tregs in CAC are required in the future [85].

## 8. Conclusions

Understanding the role of intestinal inflammation in IBD and tumorigenesis is essential for the elucidation of the mechanism(s) involved in the development of CAC. Preclinical studies have shown the importance of cytokine-related signaling in tumorigenesis (Figure 3). Clinical studies have emphasized the development of new therapeutic targets for human CAC. Understanding the immunological mechanisms of inflammation-associated tumorigenesis can help develop techniques for the prevention and therapeutic targets of CAC. Moreover, further experiments on the role of gut microbiota will help develop prophylactic treatments for CAC.

## Figures and Tables

**Figure 1 ijms-21-03062-f001:**
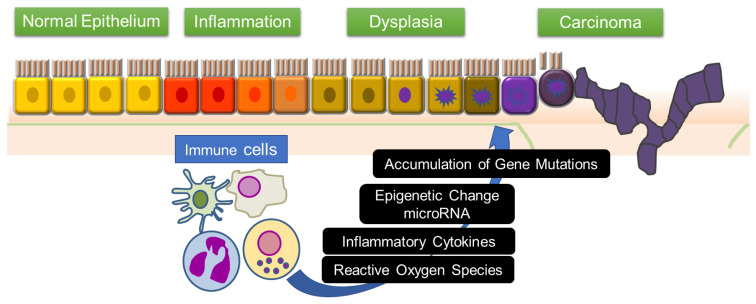
“Inflammation–dysplasia–carcinoma” sequence and immune system. In the intestinal mucosa of IBD patients, inflammation-triggering mutations and abnormalities of genes accumulate, leading to colorectal cancer via dysplasia. In this process, inflammatory cytokines and reactive oxygen species induce epigenetic changes and changes in microRNA expression in pre-malignant cells. This leads to mutations in tumor related genes, which results in the initiation of colitis-associated cancer.

**Figure 2 ijms-21-03062-f002:**
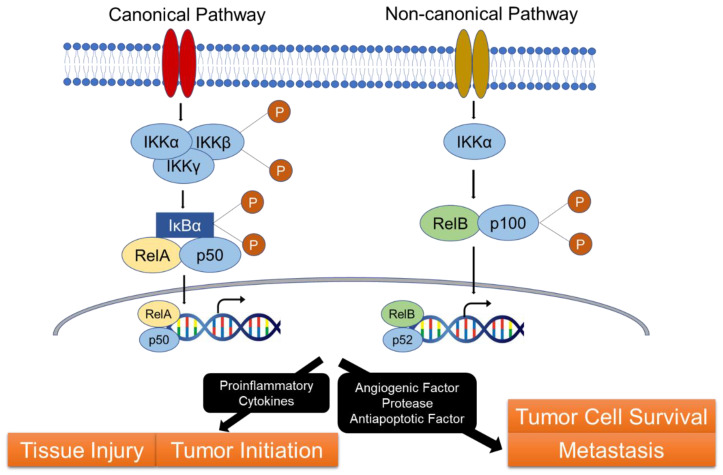
NF-κB pathway and inflammatory-associated carcinogenesis. NF-κB is involved in IBD pathogenesis. NF-κB signaling includes two pathways (the canonical and non-canonical pathways), and NF-κB forms a complex with the inhibitor IκB in the cytoplasm in both pathways. Activation of the canonical pathway results in phosphorylation of IκBα by the IκB kinase (IKK) complex. This leads translocation of RelA/p50 into the nucleus, and activation of the transcription of target genes. Activation of the non-canonical pathway leads the degradation of the p100 to p52. The p52-RelB complex translocates into the nucleus and leads the transcription of target genes. This pathway is involved in the pathogenesis of IBD. It is also involved in the carcinogenesis with cytokine-induced tissue injury, tumor initiation, and tumor progression.

**Figure 3 ijms-21-03062-f003:**
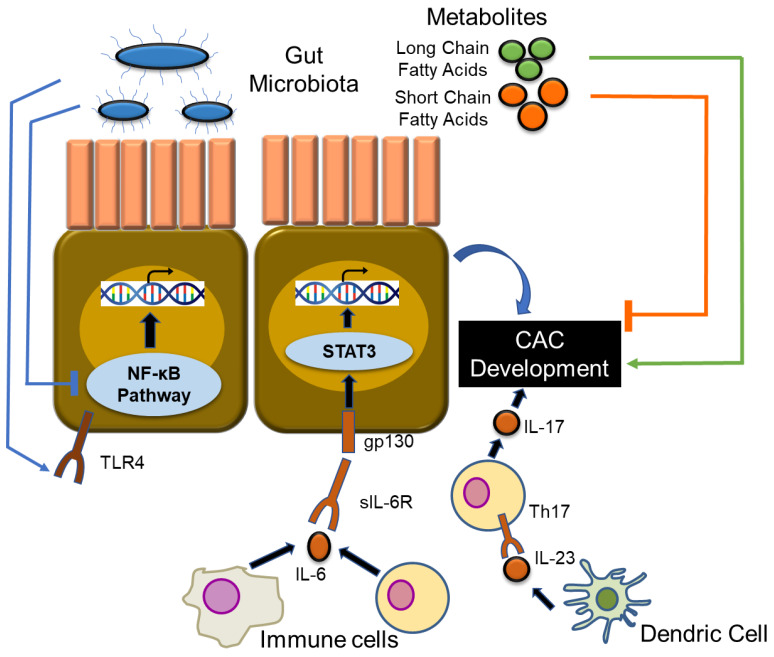
The involvement of cytokine-related signaling pathway, gut microbiome in CAC. This paper has been an overview of CAC carcinogenesis and cytokine-related signaling pathways and the gut microbiome. Various signaling pathways contribute to the development of CAC. Gut microbiota can contribute to both promotion (via activation of toll-like receptors) and suppression of the NF-κB pathway, depending on the bacterial species. Fatty acid metabolism contributions to CAC are also known. Short-chain fatty acids have a protective role in CAC carcinogenesis. In contrast, long chain fatty acids can lead to cancer development and progression. TLR4: toll-like receptor 4, IL-6R: interleukin-6 receptor.

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
