# Peer review of "Immunological Mechanisms in Inflammation-Associated Colon Carcinogenesis"

_ijms, 2020, doi:10.3390/ijms21093062_

Round 1
Reviewer 1 Report
Line 34 - 36. In addition to the aforementioned diseases, include the role that food and some of its components play in inflammatory processes with effect on CRC.
I consider that figure 1 does not represent in the best way the clonal expansion that has already formed in dysplasia, or give a description of the events during the color changes.
Author Response
Point 1: Line 34 - 36. In addition to the aforementioned diseases, include the role that food and some of its components play in inflammatory processes with effect on CRC.
Response 1: Thank you for your suggestion. We have added a brief sentence describing the correlation between dietary habits and CRC.
Point 2: I consider that figure 1 does not represent in the best way the clonal expansion that has already formed in dysplasia, or give a description of the events during the color changes.
Response 2: Thank you for pointing it out. We have added a brief sentence that describing the event occurring during color changes in figure legend.
Reviewer 2 Report
In the present manuscript, Hirano et al. provides an up-to-date critical and comprehensive overview about the main signaling pathways involved in the pathogenesis following colon inflammation to understand the underlying molecular mechanisms in CAC tumorigenesis.
MINOR POINTS
Based on the increasing interest in the field of purinergic system I suggest to add a brief paragraph describing the role purinergic receptors (P2X7 receptor) in shaping the molecular mechanisms driving intestinal inflammation toward neoplasia onset and development
Author Response
Point : Based on the increasing interest in the field of purinergic system I suggest to add a brief paragraph describing the role purinergic receptors (P2X7 receptor) in shaping the molecular mechanisms driving intestinal inflammation toward neoplasia onset and development.
Response : We appreciate for your suggestion. We added a brief chapter describing the correlation of purinergic receptors, especially p2x7 receptors, with inflammation-associated carcinogenesis.